# Defining Populations and Predicting Future Suitable Niche Space in the Geographically Disjunct, Narrowly Endemic Leafy Prairie-Clover (*Dalea foliosa*; Fabaceae)

**DOI:** 10.3390/plants13040495

**Published:** 2024-02-09

**Authors:** Ashley B. Morris, Clayton J. Visger, Skyler J. Fox, Cassandra Scalf, Sunny Fleming, Geoff Call

**Affiliations:** 1Department of Biology, Furman University, Greenville, SC 29613, USA; skylerjfox18@gmail.com; 2Independent Researcher, San Antonio, TX 78247, USA; scalf.cassandra@gmail.com; 3Department of Biological Sciences, California State University, Sacramento, CA 95819, USA; clayton.visger@csus.edu; 4Department of Biology, Georgia Southern University, Statesboro, GA 30458, USA; 5Environmental Systems Research Institute, Inc. (ESRI), Redlands, CA 92373, USA; sfleming@esri.com; 6Tennessee Ecological Services Field Office, U.S. Fish and Wildlife Service, Cookeville, TN 38501, USA; geoff_call@fws.gov

**Keywords:** conservation genetics, *Dalea foliosa*, dolomite prairies, limestone glades, microsatellites, population boundaries, suitable niche

## Abstract

Conservation actions for rare species are often based on estimates of population size and number, which are challenging to capture in natural systems. Instead, many definitions of populations rely on arbitrarily defined distances between occurrences, which is not necessarily biologically meaningful despite having utility from a conservation management perspective. Here, we introduce a case study using the narrowly endemic and highly geographically disjunct leafy prairie-clover (*Dalea foliosa*), for which we use nuclear microsatellite loci to assess the current delimitations of populations and management units across its entire known range. We model future potential suitable niche space for the species to assess how currently defined populations could fare under predicted changes in climate over the next 50 years. Our results indicate that genetic variation within the species is extremely limited, particularly so in the distal portions of its range (Illinois and Alabama). Within the core of its range (Tennessee), genetic structure is not consistent with populations as currently defined. Our models indicate that predicted suitable niche space may only marginally overlap with the geology associated with this species (limestone glades and dolomite prairies) by 2070. Additional studies are needed to evaluate the extent to which populations are ecologically adapted to local environments and what role this could play in future translocation efforts.

## 1. Introduction

Global biodiversity is threatened at an unprecedented rate by complex interactions between anthropogenic activities and climate change, but increased investment in effective conservation actions has the potential to combat the anticipated number of future species extinctions [1,2]. In the US, analyses have shown that individual species’ recovery progress over time is positively correlated with conservation funding but that the current federal budget is insufficient to address the costs associated with recovery actions across all listed species [3,4,5]. These financial limitations require those responsible for managing species recovery at both federal and state levels to make difficult decisions with respect to prioritizing which actions will both be cost-effective and have the greatest impact on the likelihood of species recovery. Perhaps the most challenging of these decisions are defining populations and characterizing the resiliency of those populations as a strategy for prioritizing recovery actions, neither of which is a trivial task [6,7,8,9,10,11]. Given these challenges and the overwhelming workload facing state and federal conservation decision makers, developing effective species conservation strategies often requires the involvement of government and non-government partners working together to develop lines of research inquiry that prioritize explicit recovery actions and implement active adaptive management strategies [12,13,14]. Here, we present a case study focused on the federally endangered plant *Dalea foliosa* (Gray) Barneby (leafy prairie-clover; Fabaceae), in which we (academic researchers and state and federal decision makers) used genetic approaches to assess the biological relevance of distance-delimited population boundaries, and we used climate models and current species distribution data to predict shifts in potential future available habitat based on current ecological requirements. We then discuss the importance of defining populations in biologically meaningful ways when instituting recovery actions.

The US Endangered Species Act of 1973 (H.R. 5961, 117th Congress) indicates that recovery plans should be developed and implemented for any species listed under the Act, and such plans should include “objective, measurable criteria” that, if met, would support a determination to remove the species from the list. These plans should also identify site-specific management actions needed for the species’ conservation and survival, as well as cost and time estimates to achieve recovery goals. While the term population is used at least 20 times in the Act, it is never defined. The United States Fish and Wildlife Service (USFWS), which is the entity charged with the implementation of the Act, has more recently developed a Species Status Assessment (SSA) Framework [11,15], which is intended to provide a scientific basis for policy application under the Act regarding species classification and recovery planning and implementation. The authors of the SSA Framework explicitly addressed the importance of and the challenges associated with defining populations, specifically highlighting methods such as genetic analysis and arbitrarily defined distances between groups as examples used by some researchers and practitioners. In the absence of other data, many state agencies charged with monitoring rare species use the arbitrarily defined distance via the concept of Element Occurrence (EO), as defined by NatureServe [16], which typically uses a default 1 km separation between units as a proxy for population delimitation. However, the authors noted that this is an inappropriate measure for many plant species, and when life history or ecological data are available to support modifications in separation distance, such changes should be considered. The issues raised here illustrate how traditional guidance on defining populations or attempts to standardize the definition of populations across plant species are not straightforward and may not be the most appropriate ways to describe biologically meaningful boundaries when attempting to allocate resources for recovery action prioritization.

Recovery actions often include the transplanting of individuals or seeds from one location to another to improve population resiliency by increasing the population size or genetic diversity or increasing redundancy by establishing new populations in perceived suitable habitat [17,18]; therefore, linking the spatial distribution of population genetic diversity on the landscape to the ecological factors regulating those populations is key to predicting potential future changes in species distributions [19,20]. Global climate change is predicted to have significant impacts on species distributions within the next 50 years, with some effects already being observed [21]. Plants are obviously incapable of migrating in the sense that animals are to escape changing environments. However, plants are capable of responding to environmental changes through either the natural selection of existing adaptive genetic variation or phenotypically plastic responses to environmental change [22]. Either of these types of responses is complex and difficult to predict without extensive experimental manipulation, but a first step in understanding how plant populations may respond to climate change is to model future changes in potentially suitable climate niches [23,24]. Such predictive models, when posited in the caveats of their limitations and coupled with population genetic studies, can be powerful tools for conservation decision makers working to prioritize recovery actions.

In this study, we present a case study of *Dalea foliosa* (leafy prairie-clover; Fabaceae), a federally endangered legume associated with the limestone glades and barrens of north Alabama and Middle Tennessee and the dolomite prairies of northern Illinois, using genetic data and future climate modeling to inform current recovery action decision making. We used nine nuclear microsatellite loci to assess genetic variation across 617 individuals from 29 different sampling locations spanning the known distribution of the species (Table 1 and Figure 1). Our analyses were performed with the goal of determining the extent to which current EO or other population boundary determinations used by managing agencies mirror the patterns observed from genetic data. Additionally, we modeled the climatic niche space of current *D. foliosa* populations, projected those models onto the forecasted climate of 2070, and overlaid these projections on top of limestone and dolomite geography to better understand where suitable climate and suitable geology intersect. In combination, we discuss the implications of the results of these two approaches for current recovery actions for this species.

## 2. Results

### 2.1. Summary Statistics

Allele frequencies by locus and sampling location are provided in Appendix A. All four Illinois populations were monomorphic for loci *Dfol082*, *Dfol092*, *Dfol241*, *Dfol016*, *Dfol171*, and *Dfol133*. Only one Illinois population was variable at locus *Dfol023* (population DF03) and at locus *Dfol003* (population DF02). Locus *Dfol005* was variable for three of the four Illinois populations, with population DF03 being the only one monomorphic at this locus. Only one Tennessee population (DF32 in the Cedars of Lebanon Complex) was monomorphic at all loci. All Alabama populations were monomorphic for loci *Dfol023*, *Dfol241*, and *Dfol171*. Three of the four Alabama populations were monomorphic for loci *Dfol005* and *Dfol133*, with the one exception in each case being population DF14. No private alleles were detected in Illinois; one private allele was detected in Tennessee (population DF24, *Dfol005*, *f* = 0.021); and four private alleles were detected in Alabama (population DF15, *Dfol092*, *f* = 0.042; DF16, one each at *Dfol082* and *Dfol016*, *f* = 0.042 and 0.083, respectively; DF17, *Dfol003*, *f* = 0.042). Significant deviation from HWE was detected for all loci except *Dfol171*. The number of populations that deviated from HWE varied by locus, ranging from *Dfol133* with 2 populations to *Dfol005* with 10 populations (Appendix A).

Summary statistics are provided in Table 2 Percent polymorphism averaged across loci varied from 0.00% (DF32) to 100% (DF23). Regionally, Illinois populations exhibited the lowest levels of polymorphism, with three of the four populations having only 11.11% polymorphic loci. In comparison, the Alabama populations ranged between 11.11% (DF15) and 44.44% (DF16 and DF17). Tennessee populations exhibited both the lowest (0.00%) and highest (100%) values, with much variation in between (Table 2). Observed heterozygosity was consistently low across Illinois populations (*H_o_* = 0.000 in DF03 to 0.023 in DF02), with Alabama populations exhibiting slightly higher values (*H_o_* = 0.009 in DF15 to 0.148 in DF14). Tennessee populations exhibited higher observed heterozygosity values than either of the other two regions, with considerable variation by population (Table 2). Consistently positive values of *F* across all three regions (but see negative values for DF01, DF25, and DF15) are indicative of inbreeding within populations.

The lowest pairwise population *F_ST_* values were between populations within the Cedars of Lebanon Complex in Tennessee (*F_ST_* = 0.008 for DF30 and DF20) and between populations within Illinois (*F_ST_* = 0.017 for DF01 and DF08). The highest pairwise population *F_ST_* values were consistently between Alabama population DF15 and populations from Tennessee (DF32, *F_ST_* = 0.981) and Illinois (DF01, *F_ST_* = 0.966). All pairwise population *F_ST_* values are provided in Appendix A. With respect to PCoA results based on the analysis of individuals, 20.64% of the observed variation was explained by the first axis, 34.72% by the first and second axes combined, and 45.66% by all three axes combined. The PCoA results for axes one and two are presented in Figure 2 These results indicate a considerable overlap among populations within the Cedars of Lebanon Complex in Tennessee; there is also a considerable overlap among populations within the Duck River Complex. Illinois populations fall within the space of Tennessee populations, primarily in the area of the Cedars of Lebanon Complex, while Alabama populations exhibit more differentiation but still overlap with Tennessee populations, primarily in the Duck River Complex. Similar but slightly different patterns were observed for PCoA based on populations (see Appendix A). There is greater differentiation between Alabama and Tennessee, although Illinois and Tennessee still show some overlap. The results of the Mantel test support isolation by distance with a *p*-value of 0.010.

### 2.2. Individual and Population Assignment

The estimated best *K* based on Evanno et al. [25] was determined to be *K* = 2, whereas the best *K* based on Pritchard et al. [26] was *K* = 22. Due to the similarity among values of *K* between 2 and 22, we chose to present the output for selected values of *K* (2, 4, 8, 12, and 22) that reflect the range of structure observed (Figure 3). Based on this output, the three geographic regions form distinct clusters (each designated with a distinct color), with no real structure observed among populations within Illinois or among populations within Alabama. Within Tennessee, similarity among populations within the Cedars of Lebanon Complex can be seen, and similarity among populations within the Duck River Complex can be seen. Other populations that are more geographically distinct appear to form distinct clusters. To further illustrate what we view to be the most biologically meaningful representation of these data, a more detailed map view of sampling locations as they relate to the STRUCTURE output for *K* = 12 is shown in Figure 4, where clusters can be clearly separated by the geographic distance between populations.

### 2.3. Current and Future Climatic Conditions

Climatic models generated for Tennessee and Illinois *D. foliosa* had area under the curve (AUC) scores of 0.997 and 0.948, respectively. The currently predicted suitable climate for both sets of *D. foliosa* populations appears to exceed (and greatly exceed in the case of Illinois) their realized distribution (Figure 5, two left panels), even when considering the intersection of suitable climate with suitable geology. The future predicated suitable climate is substantially shifted and decreased in both disjunct populations, with very little overlap projected to co-occur with suitable geology (Figure 5, two right panels).

## 3. Discussion

In the US, federally designated endangered and threatened species are subject to management considerations as defined by recovery plans, and the primary emphasis for recovery typically relates to alleviating threats to maintain or increase population resiliency and redundancy and conserve the representation of adaptive variation across a species range [11], yet the definition of population is often arbitrarily defined as a result of limited available biological information. The data we present here indicate that current population designations do not reflect the ways in which sampled sites of *D. foliosa* are connected and/or isolated from one another via gene flow, which can have considerable impacts on the allocation of limited resources for recovery actions.

In Illinois, as previously shown in Morris et al. [27], genetic variation is extremely limited both within and among populations. Based on the data presented here, the breadth of diversity observed in Illinois is fully captured within Tennessee. Tennessee is clearly both the geographic and genetic center of diversity for the species, harboring both the largest numbers of occurrences of individuals and the greatest genetic diversity within and among occurrences. Alabama, much like Illinois, exhibits relatively low levels of genetic variation but also carries private alleles relative to the other two regions. This information, combined with the results of current and future climate modeling, is used below to make explicit recommendations for defining population boundaries for recovery actions.

### 3.1. Defining Populations in Dalea foliosa

As noted, genetic variation within *D. foliosa* is exceedingly low. Edwards et al. [28] previously noted that both *D. foliosa* and a more widespread congener, *D. purpurea*, had lower levels of genetic diversity than were observed in other members of Fabaceae. McMahon and Hufford [29] also noted minimal sequence divergence among *Dalea* species in a phylogenetic analysis based on the nuclear ribosomal internal transcribed spacer (nrDNA ITS), possibly indicating that extant species are evolutionarily young. With this in mind, even subtle genetic variation within and among populations of *D. foliosa* relative to the overall level of diversity within the species may prove important for future translocation efforts.

Occurrences sampled within Illinois exhibit some of the lowest levels of diversity both within and among sites sampled across the species range, with allelic diversity so low that outcrossing between sites within Illinois is unlikely to improve genetic diversity across that portion of the range [27]. This essentially means that the Illinois sites sampled here are inferred to be functioning as a single population, with all extant occurrences found within a 16 km radius. The neighboring sites (mapped as distinct EOs) sampled here are between approximately 5 km and 20 km apart. The lack of genetic variation observed here is likely the result of a genetic bottleneck caused by Pleistocene glacial expansion and retreat, as suggested by Edwards et al. [28]. This idea of Illinois sites as a Pleistocene relict is consistent with the observation that all alleles detected in Illinois were also detected in Tennessee, suggesting that the Illinois sites emerged from a Tennessee center of diversity at some point in the evolutionary past. It is somewhat surprising that, given the hypothesized amount of time that would have passed, genetic drift or mutation has not resulted in differentiation between Illinois and Tennessee. Again, if *D. foliosa* is in fact an evolutionarily young species (which could be further tested using a time-calibrated phylogenetic analysis of *Dalea* and related genera), that could perhaps explain the patterns observed here (i.e., overall low genetic variation, resulting in even lower genetic variation in Pleistocene refugial populations). Future studies using higher-resolution genetic datasets should provide additional insight into the dynamics of these genetically depauperate sites. Based on the data presented here, without additional genetic input from other sources (i.e., Tennessee or Alabama), genetic diversity within Illinois is not expected to increase over time through natural processes, which could ultimately increase the likelihood of local extinction in the region.

In Tennessee, there are at least 20 EOs, as defined by NatureServe protocols [16], included in the present genetic dataset. Based on the combined analyses presented here, only a subset of these represent distinct genetic clusters, or populations. For example, the majority of EOs within the Cedars of Lebanon Complex appear to be functioning as a single population, with either continuous modern-day gene flow among them or a past signature of connectivity that no longer exists. According to pairwise population *F_ST_* values (Appendix A), the sampled sites within the Cedars of Lebanon Complex are as similar to each other as the sampled sites within the Illinois complex. Additionally, sites within the Cedars of Lebanon Complex are consistently more similar to each other than are other Tennessee sites within close geographic proximity to each other (e.g., Duck River Complex). The sites within the Cedars of Lebanon Complex are in relatively close proximity to each other, with distances between most neighboring sites ranging between 0.5 km and 2 km, with the greatest distance between neighboring sites being approximately 5 km. Some of the highest levels of genetic diversity observed within Tennessee are found in sites just 7 km (DF13 and DF27) and 14 km (DF23) south of the Cedars of Lebanon Complex (Table 2, Figure 3 and Figure 4). A portion of this area south of Cedars (DF13 and DF27) is known as Lane Farm, which interestingly shows some genetic similarity to Illinois sites (Appendix A). Additionally, Lane Farm shows genetic similarity to other sites across Tennessee (DF11, DF25, and DF33), which could suggest that Lane Farm represents a historic genetic source population for the species. However, we acknowledge that additional genetic resolution could change our interpretation of these results, and more data are needed to further support this idea.

As with the Cedars of Lebanon Complex, the Duck River Complex appears to form a tight genetic cluster, while the more geographically isolated sites in the same region appear to reflect distinct clusters, which could be a consequence of the geographic distance between locations there (Figure 3 and Figure 4; Appendix A). Distances between sampled sites within the Duck River Complex are quite limited (~350–800 m), with sites in the region being separated by 11 to 70 km, in contrast to distances between 550 m and 5 km between neighboring sites in the Cedars of Lebanon Complex.

In Alabama, all four sites sampled are genetically very similar to each other and exhibit very little diversity overall. Three of the four sites (DF15–DF17) are separated by approximately 0.5 km, while the fourth site (DF14) is approximately 25 km away. Based on population pairwise *F_ST_* results, there are some similarities between Alabama (DF17) and the Duck River Complex in Tennessee (DF22), as well as to those sites in and around the Lane Farm area (DF13 and DF23). Again, this could suggest that the area around Lane Farm represents a historically central source population for the broader distribution of the species. Additionally, the connectivity between the Duck River Complex and Alabama has been suggested in other glade species (see discussion below), indicating the possibility that the Duck River Complex is another important source population for the species.

Overall, based on the data presented here, Illinois is functionally a single population, as is Alabama. Both Illinois and Alabama suffer from extremely low levels of genetic diversity that warrant immediate concern. The patterns within Tennessee are more complex, with each currently recognized management complex (Cedars of Lebanon and Duck River) functioning as a single population and all other sites being sufficiently geographically isolated to largely warrant their treatment as individual populations. There is no specific distance that appears to be appropriate for defining the separation of populations. The evolutionary histories of both Illinois and Alabama mean that sites separated by as much as 20–25 km still exhibit a great deal of genetic similarity. In contrast, within Tennessee, we observed differentiation among sites within 7 km of each other. Our interpretation of the data presented here is that there is a need for connectivity corridors between geographically distant sites within Tennessee to facilitate gene flow among them and to avoid the potential for the localized extinction of isolated populations. What remains unclear is the role of pollinators in gene flow among sites, what those pollinators are, how far they can travel, and how efficient they are at successful pollen transfer. Preliminary work in Tennessee and Alabama suggests that *D. foliosa* is pollinated by a host of bee species, and these species likely forage within a mile of their nest sites (pers. comm., Bashira Chowdhury). Previous work by Molano-Flores (unpublished data) indicates that, at least in Illinois, *D. foliosa* is self-compatible, and there were no significant differences in fruit set between hand-pollinated and open-pollinated flowers. It is unclear how widespread selfing is in this species and what fitness impacts may result in subsequent generations. A greater understanding of reproductive ecology in this system would provide more insight into what constitutes evolutionarily meaningful distances between populations to facilitate gene flow.

### 3.2. Comparison with Other Calcareous Glade Endemics

To our knowledge, the only other species with a similar geographic distribution to *D. foliosa* for which genetic data have been published are *Astragalus tenneseensis* A. Gray ex Chapman (Fabaceae) [28] and *Leavenworthia stylosa* A. Gray (Brassicaceae) [30]. *Astragalus tennesseensis* is a perennial herb associated with limestone glades in north Alabama, Middle Tennessee, and dolomite glades in northern Illinois. Based on the allozyme work of Edwards et al. [28], the Illinois populations of *A. tennesseensis* were the least genetically variable, just as was observed in our study for *D. foliosa*. Additionally, the Alabama populations of *A. tennesseensis* appeared most similar to a Tennessee population they called Blue Spring, which occurs within the region we refer to as the Duck River Complex. This is notable because we observed a similar pattern in *D. foliosa*, with pairwise population *F_ST_* values between Alabama and Tennessee being most similar between our Blue Springs site (DF22) and Lane Farm (DF13; closer to the Cedars of Lebanon Complex) sites and an Alabama site (DF17).

*Leavenworthia stylosa* is an annual herbaceous species endemic to the limestone glades of the Central Basin of Tennessee. Dixon et al. [30] used nuclear microsatellite loci to test the assumptions of the abundant-center hypothesis (ACH, also known as the central-marginal hypothesis) in *L. stylosa*. The ACH has received a great deal of attention in recent years [31,32,33] due to the number of species comparisons that do not support it, as well as the various complicating factors that result in a study not meeting the assumptions of the hypothesis. Therefore, we chose not to formally test this hypothesis in our work. However, the results of Dixon et al. [30] contain several key findings that are relevant to the results we present here. First, the center of the species range for *L. stylosa* is within the Stones River watershed, which is also true for *D. foliosa*. The authors found evidence for individuals of *L. stylosa* from the same watershed sharing ancestry from the same genetic cluster, which is consistent with a hypothesis of water-based dispersal of seeds in that species. According to Baskin and Baskin [34,35], *D. foliosa* in Tennessee fruits into mid-October, at which point the shoots die and hold onto the seed through the winter. The authors described the “freshly matured seed coats” of *D. foliosa* as being impermeable to water and found that very few seeds of a cohort would germinate within the first year. In the species recovery plan [36], potential dispersal agents for *D. foliosa* were listed as wind, gravity, birds, and small mammals, with no mention of dispersal by water. If this is true, then there would be no reason that the watershed should drive genetic structure in *D. foliosa*. However, the species does tend to occur along ephemeral washes that experience localized flooding in winter to early spring, which could serve as a mechanism for dispersal, although there are no empirical studies that have documented seed dispersal mechanisms in this species. To some extent, *D. foliosa* does appear to cluster by watershed (Figure 4), although we suggest that a more viable explanation for what we observe in our data is simply a consequence of the structure by geographic distance. Locations within close geographic proximity to each other tend to exhibit shared ancestry with respect to genetic clusters. This seems likely to be a consequence of limited gene flow between more distant sites due to either pollination limitation or through limited dispersal, rather than being driven by differences in watershed. Finally, Dixon et al. [30] detected higher admixture among populations of *L. stylosa* within the Stones River watershed, concluding that this pattern was indicative of a larger influx of migrants (i.e., pollen or seeds) in that portion of the species range. We also observed greater admixture within the Stones River watershed for *D. foliosa*, specifically within the Cedars of Lebanon Complex (Figure 3 and Figure 4). It is notable that the Cedars of Lebanon Complex (~9000 acres of protected land) is considered the largest undeveloped contiguous area of Central Basin limestone glades and barrens in the world. Additionally, this complex harbors the largest number and highest density of occurrences of *D. foliosa*. Given these facts and the potential for spatial connectivity among extant sites, it should be no surprise that we observe the greatest genetic admixture in this portion of the species range.

### 3.3. Predicted Future Suitable Climatic Conditions for Dalea foliosa

As a narrow endemic, ecological models of *Dalea foliosa* must come with a substantial caveat. Specifically, *D. foliosa* likely has very specific microhabitat requirements, which could include shallow topsoil and flooding frequency [35,36,37], that are not reflected in broad global models. Therefore, the models we discuss here are better interpreted as predicted suitable climatic conditions and their intersection with the limestone and dolomite geology that defines the glades this species inhabits. Based on both the present and predicted future climatic models, it appears that by 2070, following a moderate climate change scenario, limestone/dolomite geology will only intersect with a modest region of climate similar to the current Illinois *D. foliosa* experience [38,39]. Conversely, in 2070, very little area is predicted to have climatic conditions similar to the current Tennessee *D. foliosa* population conditions. Are there substantive physiological differences between Tennessee and Illinois *D. foliosa* that led to them inhabiting climatically divergent regions? Or are there little to no physiological differences between the two regions and the climatic difference is indicative of a broader climatic tolerance that reflects the pre-disjunction range? Future work involving either reciprocal transplants and/or common garden experiments will be critical for understanding the actual impact climate change will impose on the realized niche space of *D. foliosa*. If Illinois populations of *D. foliosa* are empirically shown to differ in climatic preference from Tennessee or Alabama populations and are better suited to survive the forecasted climatic changes, the limited genetic diversity of these populations may become an area where management officials take a more active role in cultivating genetic diversity via transplants. Furthermore, active management to prevent the encroachment of woody vegetation within extant and potentially suitable *D. foliosa* sites will be needed in order to conserve existing genetic variation and promote gene flow among sites.

## 4. Methods

### 4.1. Study Species

*Dalea foliosa* is a short-lived perennial associated with the highly fragmented limestone glades and barrens of northwest Alabama and Middle Tennessee and the equally imperiled dolomite prairies of northern Illinois. The species was federally listed as endangered in 1991, and at that time, there were 29 “known populations” in three states: Alabama (2), Illinois (3), and Tennessee (24) [40]; recovery criteria indicated that the species could be “considered recovered and eligible for delisting when at least 3 high-viability populations in each Illinois and Alabama and 12 high-viability populations in Tennessee are protected and managed”. The primary threat to the species is competition from woody encroachment under restricted fire regimes, followed by habitat destruction and anthropogenic development throughout its geographic range. Furthermore, Tennessee is considered the center of its geographic distribution and the “reservoir of genetic diversity”, leading to a recommendation to conserve as many Tennessee populations as possible. All sites in the species range included here are mapped as EOs within Biotics, a national data management platform from NatureServe (pers. comm., Cathy Pollack [USFWS–Illinois], Caitlin Elam [Tennessee Department of Environment and Conservation (TDEC) Division of Natural Areas (DNA)], and Wayne Barger [Alabama Department of Conservation and Natural Resources (ALDCNR)]). The earliest possible recovery date was estimated to be 2005 [40]; as of the most recent 5-Year Review [41], none of the recovery criteria had been met.

Previous genetic work by Edwards et al. [28] surveyed allozyme diversity (nine enzyme systems) among 240 individuals from 10 populations (3 in Illinois, 6 in Tennessee, and 1 in Alabama) across the range of the species. They identified lower-than-expected levels of isozyme diversity, with the Tennessee populations exhibiting the highest levels of variation. Furthermore, the authors concluded that the observed genetic patterns were most likely a result of past evolutionary history related to glacial expansion and retreat rather than to modern-day population dynamics or the genetic makeup of the soil seed bank. More recently, members of our team surveyed nuclear microsatellite diversity at six loci for 226 individuals from 11 populations (9 from Illinois and 2 from Tennessee) to assess genetic diversity within and among managed Illinois sites to evaluate the impacts of augmentation and introduction efforts [27]. As observed by Edwards et al. [28], our results indicated that there are extremely low levels of genetic diversity among Illinois populations, with Tennessee populations exhibiting much higher levels of diversity. Furthermore, we observed less genetic variation within and among the nine Illinois sites surveyed than within one of the Tennessee sites surveyed. We concluded that the Illinois soil seed bank likely has insufficient genetic variation to rescue the current populations but that additional understanding of ecophysiology is needed before we can consider the translocation of material from Tennessee as a possible recovery action.

### 4.2. Field Sampling

The leaf material for this study was primarily sampled between 2014 and 2022, with the majority of sites being sampled in 2014 and 2015. In the present study, we included a total of 29 sites from across the species range, including 4 in Illinois, 21 in Tennessee, and 4 in Alabama (Table 1). All sites are thought to be naturally occurring, with the exception of one site in Illinois (Keepataw), which is naturally occurring but was augmented with seedlings sourced from other Illinois sites (see [27]). All Illinois sites included here are within an approximately 16 km radius and are mapped as separate EOs; Tennessee sites occur within an approximately 45 km radius and are each mapped as separate EOs; and in Alabama, three of the four sites are within 800 m of each other, with the fourth site being approximately 25 km away, and each of the four is mapped as a separate EO. The southernmost Illinois site is approximately 590 km from the northernmost Tennessee site, while the southernmost Tennessee site is approximately 130 km from the northernmost Alabama site (Figure 1). In Tennessee, the Tennessee Division of Natural Areas (DNA) within the Tennessee Department of Environment and Conservation (TDEC) defines two distinct management units: the Cedars of Lebanon Complex and the Duck River Complex. The Cedars Complex includes eight sites from the current study, while the Duck River Complex includes three sites from the current study, all of which are indicated in Table 1. Note that these management unit designations were not designated as such to imply any biological relationship; instead, they represent EOs in close geographic proximity to each other that are a contiguous state-owned management unit (pers. comm., Caitlin Elam). The remaining sites are treated as separate management units. The GPS coordinates of sites across the range are not provided here for the protection of the species. All material was collected by staff working with USFWS, which is responsible for the monitoring of the species. A leaf or several leaflets were collected from each of a maximum of 30 individuals at each site and immediately stored in silica-gel desiccant for DNA preservation.

### 4.3. Microsatellite Development and Genotyping

Total genomic DNA was extracted from leaf material using either the Qiagen DNeasy Plant Mini Kit (Qiagen, Valencia, CA, USA) or the Qiagen DNeasy Plant Pro Kit following the manufacturer’s protocols with minor modifications; recommendations for difficult species with high concentrations of polyphenolic compounds were followed. Seven nuclear microsatellite loci previously developed for *D. foliosa* [27], as well as three previously unpublished loci, were selected for the present study (Table 3). Each locus was amplified individually following the protocols outlined in Morris et al. [42]. We used the three-primer approach of Schuelke [43] to fluorescently label PCR products. For each locus, a 17-base tail (5′-GTA AAACGACGGCCAGT-3′) was added to the 5′ end of each forward primer, and a 7-base “pigtail” (5′-GTTTCTT-3′) was added to the 5′ end of each reverse primer. A third primer was designed to match the 17-base tail and was fluorescently labeled with either FAM, VIC, or NED fluorophores. The final concentrations for each reaction included 1X Platinum Taq buffer (Life Technologies, Foster City, CA, USA), 2 mM MgCl_2_, 0.5 μM forward primer with the 5′-M13 tail, 0.15 μM fluorescently labeled M13 primer, 0.2 μM pig-tailed reverse primer, 0.2 mM dNTPs, 0.5 U Platinum Taq (Life Technologies), and 1 μL of DNA. PCR reactions were performed in 96-well plates, with approximately 1/3 of each plate containing positive controls to facilitate consistency in the sizing of alleles across genotyping runs. Two to four loci were poolplexed for genotyping using the LIZ-500 size standard on an ABI 3130 (Life Technologies, Foster City, CA, USA) either at Cornell University Institute of Biotechnology or in the Department of Biology at Middle Tennessee State University. Alleles were sized using GeneMarker MTP software v. 2.6.0 (Softgenetics LLC, State College, PA, USA).

### 4.4. Summary Statistics

As previously noted, approximately 1/3 of samples were replicated across genotyping runs to monitor consistency in genotyping calls. Through this process, we identified inconsistencies in run patterns for one locus (*Dfol020*) to the extent that we chose not to retain this locus in the dataset. All subsequent analyses were based on a reduced, nine-locus dataset (Table 3). Genetic diversity metrics were calculated using GenAlEx 6.5 [44,45]. Summary statistics were calculated for each population averaged over all loci as well as by individual locus. Loci were tested for Hardy Weinberg Equilibrium, and summary statistics included percent polymorphic loci (*%P*), the mean number of alleles (*N_a_*), the mean number of effective alleles (*N_e_*), the mean number of private alleles (*P_A_*), observed heterozygosity (*H_o_*), unbiased expected heterozygosity (*uH_e_*), and the fixation index (*F*). *F* values near zero indicate random mating; positive values (up to a value of 1.00) indicate inbreeding or undetected null alleles; and negative values indicate heterozygote excess as a result of either negative assortative mating or heterotic selection [44]. Pairwise population *F*_ST_ values were calculated, and Principal Coordinate Analysis (PCoA) was performed on genetic distance matrices for both individuals and populations, as calculated in GenAlEx following the methods of Peakall et al. [46] and Smouse and Peakall [47], using the covariance-standardized method to further characterize structure within and among populations. A Mantel test for isolation by distance was performed across all populations using a pairwise geographic distance matrix and the output from the population pairwise *F_ST_* analysis described above, using 99 permutations.

**Table 3 plants-13-00495-t003:** Characterization of nuclear microsatellite loci developed for the federally endangered *Dalea foliosa* (leafy prairie-clover; Fabaceae).

Locus	Primer Sequences (5′-3′)	Repeat Motif	Allele Size (bp)
*Dfol003*	F: GACATGGGTGGGTATGATTGAAG R: CGCGTGATGAGACCCTTATAAAG	(AG)_8_	260
*Dfol005*	F: ATGAAGGAAGATAATACCCGGCC R: CTTGCTGCTTTCGAATCATTCAC	(AG)_13_	164
*Dfol016*	F: CACACAAACAGGAAGAGAGATGG R: AACTAATGATTCCACCAGCCAAC	(AG)_10_	203
*Dfol020* ^1^	F: TCAGCGTCTTTGATCATCTGTTC R: TTTCAGGGTGTTTGACAAGGATC	(AT)_15_	255
*Dfol023*	F: ACCGATGATAGAAGAAAGCAAGG R: TGCTTTCATAGTCTTCAACGTCC	(AAC)_7_	194
*Dfol082* ^2^	F: CACACAAACAGGAAGAGAGATGG R: AACTAATGATTCCACCAGCCAAC	(AG)_8_	201
*Dfol092*	F: TTTCGCATCGTAACCTGAAGAAG R: GTCTCTGTGTCCTTCATTCCTTG	(AGG)_7_	213
*Dfol133* ^2^	F: CACACCGTGGAATCTTACTGTG R: ACCCTCCTTTCCACAAACAATAAG	(AC)_8_	190
*Dfol171* ^2^	F: TTCTTCACCTGCGTTGATTATGG R: ATCCAGCAAAGTCTATGAAGCTG	(AG)_9_	251
*Dfol241*	F: TGTGACACAAGTTGAACAAGATC R: AGAAATCGCTGTTTCCTTCCAAC	(AAG)_6_	173

^1^ Locus *Dfol020* was removed from the analysis due to inconsistent performance. See text for details. ^2^ Previously unpublished loci; all other loci previously described in Morris et al. [27].

### 4.5. Individual and Population Assignment

Individuals were assigned to genetic clusters using STRUCTURE 2.3.4 [26,48] running 150,000 Markov chain Monte Carlo (MCMC) replicates following a burn-in period of 50,000, admixture assumed, and values of *K* ranging from 1 to 29, with 40 replicates per *K* value, all using STRUCTURE ver. 2.3.4 [26,49] as implemented in ParallelStructure [50] on the CIPRES Science Gateway [51]. The output was then submitted to CLUMPAK [52] to estimate the best value of *K* using both Evanno et al. [25] and Pritchard et al. [26]. According to Funk et al. [53], the Pritchard et al. method was more consistent at predicting known horse breeds from a large dataset than was the Evanno method, leading us to compare the results of both methods here. The visual output for *K* was generated using the CLUMPAK main pipeline with default settings for CLUMPP [54] and DISTRUCT [55].

### 4.6. Current and Future Climatic Conditions

Occurrence data for Tennessee and Illinois *D. foliosa* populations (*n* = 82 and *n* = 9, respectively) were acquired from the state and federal agencies responsible for the annual monitoring of the species. Climatic niche modeling was not conducted on Alabama *D. foliosa* due to the low number of populations and their close proximity to one another. Current and future (2071-2100 ipsl-cm6a-lr ssp370) climatic layers based on those used in Morris et al. [27] were obtained from CHELSA at a 30-arc-second resolution (Bio 10—mean air temp of the warmest quarter; Bio 11–mean air temp of the coldest quarter; Bio 16—mean monthly precipitation of the wettest quarter; Bio 17—mean monthly precipitation of the driest quarter; gst—mean temperature of all growing season days based on TREELIM; and gsp—precipitation sum accumulated on all days during the growing season based on TREELIM) [56,57]. A polygon mask including only the Eastern United States where *D. foliosa* occurs was created using QGIS ver. 3.30.3 [58] and used to trim GIS layers in R ver. 4.3.2 [59].

The logistic output of MaxEnt ver. 3.4.4 [60,61] was used to generate climatic niche models. In addition to default settings, a 15% random test percentage and a maximum of 5000 iterations were used to generate the current model, which was then projected on the future climate dataset. The resulting current and future distributions were visualized in QGIS [58] with a base map plus a GIS layer illustrating the distribution of limestone and dolomite obtained from the State Geologic Map Compilation [62].

## 5. Conclusions

*Dalea foliosa* has an unexpectedly low level of genetic variation for a species with such a broad geographic distribution. Additionally, the unique association of this species with specific geological substrates, coupled with the observed modern-day geographic disjunction, raises challenging questions for conservation biologists and land managers: Are the observed patterns innate to the pre-human evolutionary history of the species? Or are they the result of human impacts within the last several hundred years? The current state of the species is most likely a combination of these two scenarios, and recovery actions should consider these histories moving forward. As discussed above, there is no clear geographic distance that can be used to delineate populations of this species. While it is likely valuable to those who are responsible for annual demographic monitoring to define EOs, these should not be considered biologically meaningful units in *D. foliosa*. We note that while we did not discuss the population viability index (PVI) here, this tool is being used as a way to assess recovery in this species [39]. There are many ways in which population viability can be evaluated through models [9,63,64,65], such that a careful re-evaluation of the strategy used to assess viability in *D. foliosa* may be warranted. In particular, the incorporation of genetics into population viability analysis (PVA) models could be a valuable exercise. Given the potential for individual plants to live at least eight years, the genetic monitoring of populations on a 10-year cycle using higher-resolution genetic datasets will provide an assessment of the level of genetic turnover within populations, which would inform long-term strategies regarding the potential for recovery of the species leveraging the soil seed bank (if possible).

It is unclear whether the low levels of neutral genetic variation observed in the present study will have a significant impact on long-term fitness within *D. foliosa* populations. Furthermore, our future predictions of suitable climatic conditions for the species rely on an assumption that the ecophysiological range of the species is fairly stable, with limited plasticity under future predicted changes in climate. To assess concerns related to fitness and physiological plasticity, future studies should focus on adaptive genetic variation in both common garden experiments and controlled experimental settings. Such studies will provide additional insight into the factors most likely to drive or inhibit germination and reproductive output under modified environmental stresses. Recovery in *D. foliosa* will depend on a coordinated, collaborative effort among partners to better understand the ecological determinants of evolutionary success in this species.

## Figures and Tables

**Figure 1 plants-13-00495-f001:**
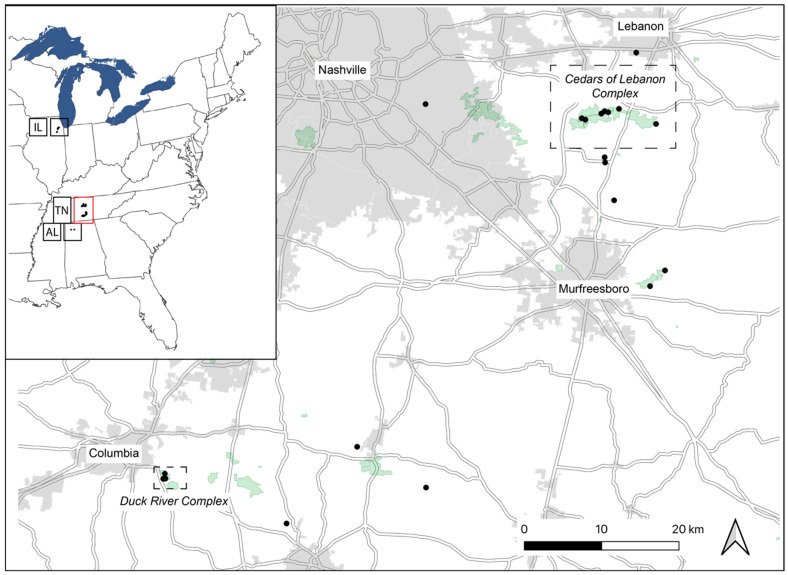
Localities of *Dalea foliosa* genotyped for nine nuclear microsatellite loci in the present study. The inset shows the known species range, with black dots indicating sampling localities; disjunct regions are defined using standard state codes: AL = Alabama; TN = Tennessee; IL = Illinois. The red box in the inset represents the area that is shown in the larger map. Again, black dots indicate sampling localities, and two areas of focus for current recovery actions in Tennessee (Cedars of Lebanon Complex and Duck River Complex) are delineated by dotted lines. Details for each sampled locality are provided in Table 1.

**Figure 2 plants-13-00495-f002:**
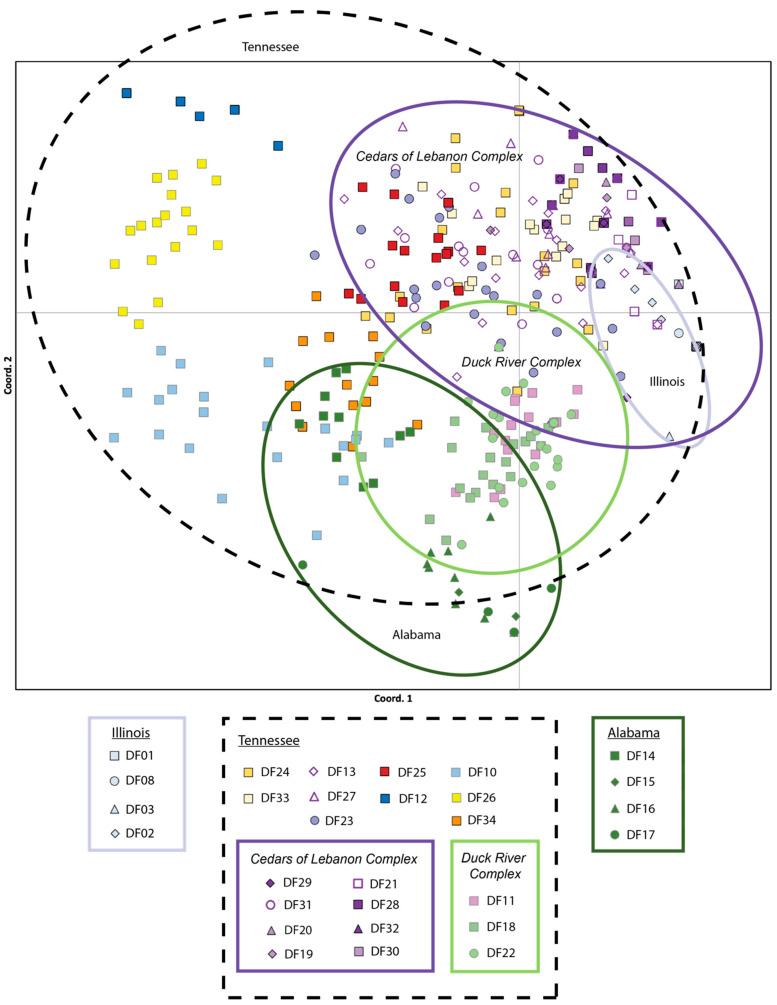
Principal Coordinate Analysis (PCoA) of the 29 populations of *Dalea foliosa* sampled across the species range and genotyped for nine nuclear microsatellite loci. The percentage of variation explained by the first and second axes combined was 34.72%. Color coding is consistent with the STRUCTURE clusters presented in Figure 3, where Illinois sites are coded in pale blue, Tennessee sites are encircled with a black dotted line and further grouped by management unit (Cedars of Lebanon Complex is encircled in purple, Duck River Complex in light green), and Alabama sites are coded in dark green.

**Figure 3 plants-13-00495-f003:**
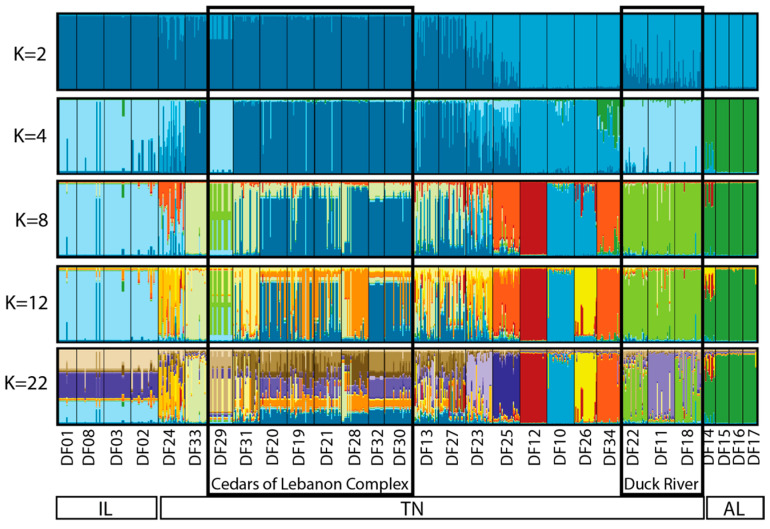
STRUCTURE analysis of the 29 sampled sites of *Dalea foliosa* for the nine microsatellite loci included in the present study. The best approximation of *K* using Evanno et al. [25] was *K* = 2; the best approximation of *K* using Pritchard et al. [26] was *K* = 22. Values of *K* between 2 and 22 presented here are those that illustrate the greatest change between values within that range. Sampling locations from left to right are consistent with the order of sites from north to south, as listed in Table 1 and Table 3.

**Figure 4 plants-13-00495-f004:**
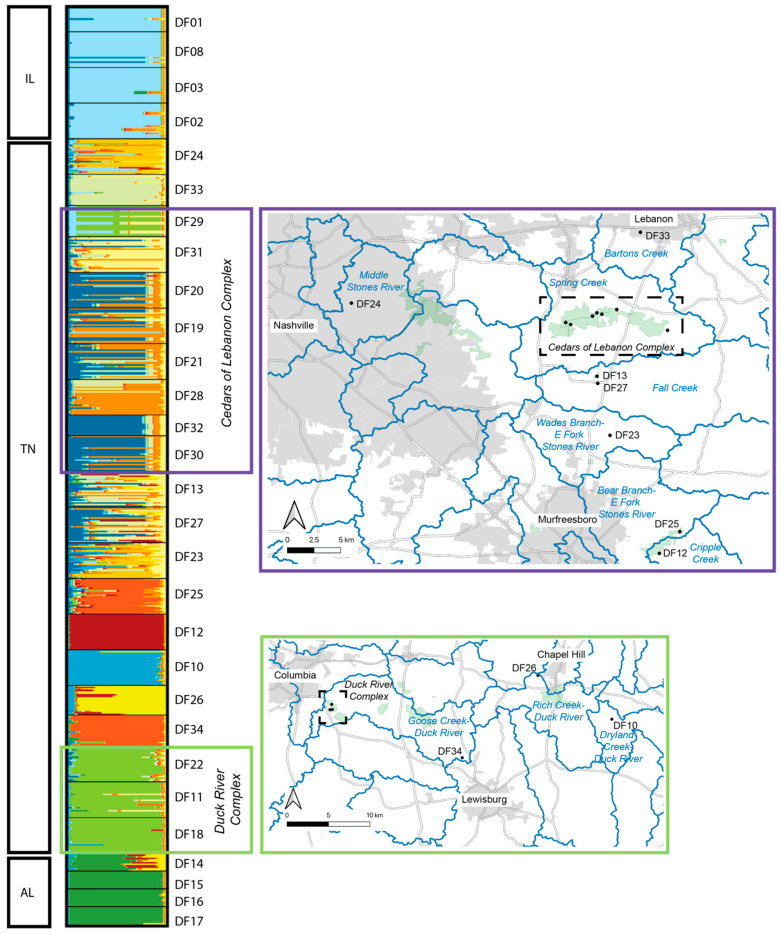
STRUCTURE analysis of *Dalea foliosa* localities sampled in Tennessee for the present study. The two maps presented here provide details for each of the two managed complexes (Cedars of Lebanon Complex and Duck River Complex) shown in Figure 1. The STRUCTURE output here is for *K =* 12, with each value of *K* represented as a different color. Within the maps, HUC-12 watersheds are delineated by blue lines.

**Figure 5 plants-13-00495-f005:**
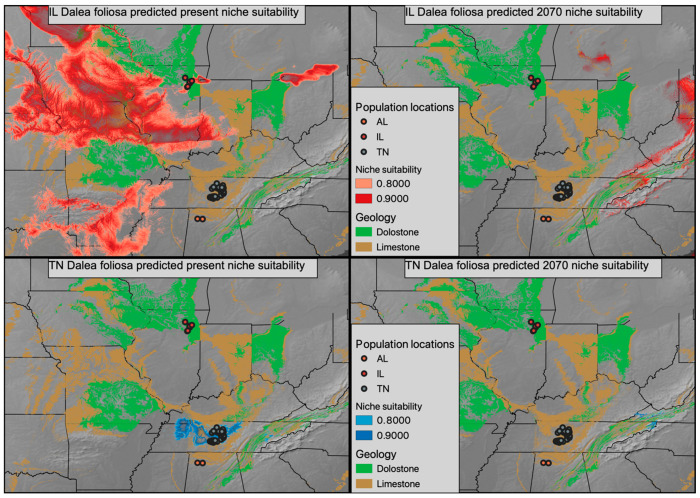
Present niche suitability for *D. foliosa* predicted using Illinois populations (**top left panel**) and Tennessee populations (**bottom left panel**) in red and blue, respectively. The climate models of the Illinois and Tennessee populations were then projected onto the forecasted climate of 2070 in red (**top right panel**) and blue (**bottom right panel**), respectively. A geologic base layer was used to represent the occurrence of dolostone and limestone, the preferred geologies of *D. foliosa*.

**Table 1 plants-13-00495-t001:** *Dalea foliosa* localities sampled for the present study.

Site Name		Site Code	Month Sampled	Collector(s)
*Dalea foliosa sites sampled in Illinois*				
Keepataw		DF01	July 2014	Pollack
Material Services		DF08	Aug 2014	Pollack
Dellwood		DF03	July 2014	Pollack
Midewin		DF02	Aug 2014	Pollack

*Dalea foliosa sites sampled in Tennessee*				
Hamilton Creek		DF24	Aug 2015	Fleming
Lebanon		DF33	2022	Elam
Cedar Forest Rd W	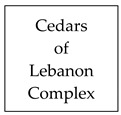	DF29	June 2021	Elam, Fox, Morris
Vesta		DF31	June 2021	Elam, Fox, Morris
Rowland Barrens E		DF20	Sept 2015	Bishop, Fleming, Williams
Rowland Barrens W		DF19	Sept 2015	Bishop, Fleming, Williams
Richmond Shop Rd		DF21	Sept 2015	Fleming
Cedar Forest Rd S		DF28	June 2021	Elam, Fox, Morris
Cedars Powerline		DF32	June 2021	Elam, Fox, Morris
Hidden Springs		DF30	June 2021	Elam, Fox, Morris
Lane Farm		DF13	Aug 2015	Bishop, Williams
HWY 452 (Lane Farm)		DF27	Sept 2015	
Holly Grove Rd		DF23	Sept 2015	Bishop, Fleming
Hall Farm		DF25	Aug 2015	Bishop, Crabtree
Flat Rock		DF12	2015	
Burnt Hill Rd		DF10	Sept 2015	Bishop, Williams, Call
TVA Powerline (Chapel Hill)		DF26	Sept 2015	Bishop, Fleming
Berlin Glade		DF34	Aug 2022	Elam, Call, Cogburn
Blue Springs	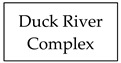	DF22	Sept 2015	Bishop, Fleming
Columbia Glade W		DF11	Sept 2015	Fleming
Columbia Glade E		DF18	Sept 2015	Bishop, Williams

*Dalea foliosa sites sampled in Alabama*				
Franklin Co., AL		DF14	July 2015	Barger
Lawrence Co., AL TVA ROW		DF15	July 2015	Barger
Lawrence Co., AL Paved roadside		DF16	July 2015	Barger
Lawrence Co., AL Dirt roadside		DF17	July 2015	Barger

**Table 2 plants-13-00495-t002:** Summary statistics for nine microsatellite loci over 617 individuals across 29 sites sampled for *Dalea foliosa*.

Site Name	*N*	*%P*	*N_a_*	*N_e_*	*H_o_*	*H_e_*	*F*		
*Illinois sampling locations*									
DF01	16	11.11	1.111	1.007	0.007	0.007	−0.032		
DF08	24	11.11	1.222	1.026	0.005	0.021	0.781		
DF03	24	11.11	1.111	1.020	0.000	0.017	1.000		
DF02	24	22.22	1.222	1.091	0.023	0.065	0.632		

*Tennessee sampling locations*									
DF24	23	66.67	2.444	1.708	0.219	0.330	0.400		
DF33	20	44.44	1.556	1.314	0.106	0.144	0.252		
DF29	21	11.11	1.111	1.077	0.000	0.045	1.000	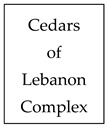	
DF31	21	44.44	1.444	1.283	0.101	0.158	0.285		
DF20	23	44.44	1.444	1.057	0.033	0.047	0.127		
DF19	24	55.56	1.556	1.234	0.046	0.138	0.744		
DF21	24	33.33	1.444	1.109	0.056	0.077	0.455		
DF28	23	22.22	1.333	1.269	0.108	0.119	0.080		
DF32	12	0.00	1.000	1.000	0.000	0.000			
DF30	23	33.33	1.333	1.052	0.037	0.040	0.019		
DF13	22	88.89	2.333	1.389	0.147	0.250	0.417		
DF27	23	66.67	2.000	1.316	0.044	0.201	0.764		
DF23	23	100.00	2.333	1.640	0.235	0.363	0.412		
DF25	23	44.44	1.667	1.229	0.130	0.132	−0.021		
DF12	23	11.11	1.111	1.107	0.010	0.055	0.823		
DF10	21	55.56	1.667	1.396	0.119	0.214	0.359		
DF26	19	66.67	2.000	1.412	0.163	0.243	0.373		
DF34	21	44.44	1.444	1.310	0.069	0.175	0.580		
DF22	23	44.44	1.556	1.386	0.169	0.206	0.185	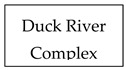	
DF11	22	22.22	1.222	1.205	0.087	0.107	0.188		
DF18	23	55.56	1.778	1.344	0.110	0.182	0.424		

*Alabama sampling locations*									
DF14	11	33.33	1.667	1.546	0.148	0.206	0.286		
DF15	11	11.11	1.111	1.010	0.009	0.009	−0.043		
DF16	11	44.44	1.556	1.184	0.058	0.114	0.267		
DF17	12	44.44	1.556	1.092	0.028	0.076	0.635		

*N* = number of individuals sampled and genotyped; *%P* = percentage of polymorphic loci; *N_a_* = mean number of alleles per locus; *N_e_* = number of effective alleles per locus, calculated as 1/(Σp_i_^2^); *H_o_* = observed heterozygosity, calculated as number of heterozygotes/*N*; *H_e_* = expected heterozygosity, calculated as 1 − Σp_i_^2^; *F* = fixation index, calculated as *H_e_* − *H_o_*/*H_e_*.

## Data Availability

Data are available from the corresponding author upon request. The data are not publicly available due to sensitive locality information.

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
