# Peer review of "Defining Populations and Predicting Future Suitable Niche Space in the Geographically Disjunct, Narrowly Endemic Leafy Prairie-Clover (*Dalea foliosa*; Fabaceae)"

_plants, 2024, doi:10.3390/plants13040495_

Round 1

Reviewer 1 Report

Comments and Suggestions for Authors

I have enjoyed reviewing Morris et al. paper titled, 'Defining population and predicting future suitable niche space in the geographically disjunct, narrowly endemic leafy prairie-clover.' The introduction is well-written and addresses significant issues in assessing Endangered and Threatened Species. I agree that EO is not a meaningful biological unit. The current ESA guidelines are based on geographic parameters related to wind-pollinated tree species (not scale-appropriate for insect-pollinated forbs). Please remove the extra 'to' in line 54. The objectives of this study were clearly articulated and the scope of the genetic sampling is sufficient to address the stated goals.

The molecular methods are appropriate for this study; however, the authors must fix the format for Tables 1 & 3. Population genetic data analysis should be straightforward, and most of what the authors present is just that. Choosing not to analyze these data by EO or natural area where they were sampled, the authors could not test for deviation from HWE among sites. Testing for deviations among EO or sites is informative for establishing conservation genetic best management practices (genetic rescue). I have concerns about using individual genotypes in the PCoA rather than using EO allele frequency data to analyze population genetic spatial relationships, especially given the author's previous work. Running the PCoA with 26 EO 'populations' would be more informative than individual genotypes. More to the point, the authors used the Bayesian Analysis STRUCTURE to assess clusters (groupings, populations) and noted two different procedures to determine the number of clusters. Evanno et al. is the standard method (cited almost 23,000 times), while Pritchard et al. is based on horse breeds. The Evanno et al. method returned two groups, while the Pritchard et al. approach indicated 22 groups. The authors did not use either of these grouping patterns, opting for 12 groups because it seems to follow geographic patterns better (Figure 4). Why did they do the analyses only not to use the results? The authors finally came around to essentially four groups in their discussion, but this was not consistent with their STRUCTURE analyses.

·         Rerun the PCoA based on 26 EO and see if the separations among sampling locations follow with 2 or 3 axes.

·         You have likely already run tests for deviations from HWE, based on the summary stats presented in Table 3. These results may be more supportive of what you suggest in the discussion.

The niche suitability analyses are instructive and what one would expect. I concur with the authors about establishing gene flow corridors to promote genetic diversity within the core of D. foliosa distribution. The results of this study could also be used to justify increasing genetic diversity in the Illinois and Alabama edge-of-range populations. Nature selection acts upon phenotypes, which are the product of the genotype by environmental interactions. Genetic bottleneck or founder effects seem like a more plausible explanation for low genetic variation in the Illinois and Alabama populations than ecotypic variation. An outcrossing–genetic rescue study would be informative for the natural resource managers monitoring these populations. Dalea purpurea has a mixed mating system; however, there is a significant reduction in fitness with selfed offspring. I wonder if D. foliosa is similar?

·         Advocate for data-driven management decisions. EO may be useful for documenting endangered species; however, the results of this study would suggest viewing the EOs within a metapopulation framework.

Reviewer 2 Report

Comments and Suggestions for Authors

This paper describes the use of population genetic structure to define populations and management units for the purposes of accomplishing recovery objectives when demands on resources are high, and time and funding are limited. The authors seek to test the biological relevance of defining populations based on distance or element occurrences and also to provide specific information to improve management of the endangered plant Dalea foliosa.  They also examine identified management units in the context of climate change and predicted future distributions. This paper contributes to our general understanding of population genetic structure in rare plants with restricted distributions as well as our understanding of genetic structure in this region of the US. Further, it demonstrates the utility and importance of genetic information in the management of rare and endangered plants. The paper is extremely well-written and clearly presented, with strong attention to detail. The field and lab work has been carefully conducted and analyses are correctly applied and interpreted (for the most part). I have a few minor concerns I will list below.

Some aspects of Table 1 are unclear (possibly due to formatting resulting from automated creation of the manuscript for review). Listing the state name within the same column as the population names is easily overlooked and somewhat unclear; perhaps a separate column for state would be better. The purpose of the horizontal black lines on occasion throughout the table is not clear, and the boxes with the two complex names don’t stay in the correct place when reformatted for review. 

Some additional information on climatic niche modeling  methods in section 2.6 and in the discussion in general would be useful. Why were the specific bioclimatic variables chosen? A brief mention of how these were selected would be good. Why were climatic layers clipped to such a broad area of the eastern US? Could this have an effect on the overly broad predicted distribution for Illinois populations? I have some concerns with the very large predicted niche suitability for IL populations, and no possible explanations for this pattern are given in the discussion. The shifts in niche suitability in IL and TN in 2070 are also quite dramatic. Does this fit with general expectations of plant ranges under predicted climate change? The discussion of results from the niche modeling in section 4.3 is limited and could be improved.

Some additional discussion of the various values for “K” in structure analyses would be useful vs. just showing us several of them. These results are relevant to the conclusions drawn for population delineation and management. Is genetic variation so low that it affects our ability to detect structure? Which of these values for K makes the most sense biologically? Why was K=12 chosen for figure 4? Is K=22 meaningful at all? The horizontal lines shown in K=22 for the Cedars of Lebanon Complex indicate a real lack of structure in this particular model.

Clearly geography and geographic distance play an important role in genetic structure in this system. Why was a test for isolation by distance not performed, even if just on an appropriate subset of the populations?

Line 214 – needs a numbered reference rather than a dated one.

Line 229 – methodology needs additional details. PCoA was performed on what? Genetic distances? Which distance measure?

Line 372  - “Foliosa” is capitalized
